# Insights on the Functional Role of Beta-Glucans in Fungal Immunity Using Receptor-Deficient Mouse Models

**DOI:** 10.3390/ijms22094778

**Published:** 2021-04-30

**Authors:** Mark Joseph Maranan Desamero, Soo-Hyun Chung, Shigeru Kakuta

**Affiliations:** 1Laboratory of Biomedical Science, Graduate School of Agricultural and Life Sciences, The University of Tokyo, 1-1-1 Yayoi, Bunkyo-ku, Tokyo 113-8657, Japan; mmdesamero@up.edu.ph; 2Department of Basic Veterinary Sciences, College of Veterinary Medicine, University of the Philippines Los Baños, Laguna 4031, Philippines; 3Division of Experimental Animal Immunology, Research Institute for Biomedical Sciences, Tokyo University of Science, 2669 Yamazaki, Noda, Chiba 278-0022, Japan; shchung@rs.tus.ac.jp

**Keywords:** beta-glucan, dectin-1, gene-modified mice, CR3, EphA2, PRR

## Abstract

Understanding the host anti-fungal immunity induced by beta-glucan has been one of the most challenging conundrums in the field of biomedical research. During the last couple of decades, insights on the role of beta-glucan in fungal disease progression, susceptibility, and resistance have been greatly augmented through the utility of various beta-glucan cognate receptor-deficient mouse models. Analysis of dectin-1 knockout mice has clarified the downstream signaling pathways and adaptive effector responses triggered by beta-glucan in anti-fungal immunity. On the other hand, assessment of CR3-deficient mice has elucidated the compelling action of beta-glucans in neutrophil-mediated fungal clearance, and the investigation of EphA2-deficient mice has highlighted its novel involvement in host sensing and defense to oral mucosal fungal infection. Based on these accounts, this review focuses on the recent discoveries made by these gene-targeted mice in beta-glucan research with particular emphasis on the multifaceted aspects of fungal immunity.

## 1. Introduction

Throughout the past decades, studies that focus on the carbohydrate-rich chemical compounds that are elaborated by food components and organisms have become a mainstay of intense scientific pursuits. Notably, beta-glucans, defined by a backbone of β-(1,3)-D-glucose with interspersing side chains made up of β-(1,6) glycosidic linkage, have commanded great attention due to their broad biofunctional activities including anti-infective [1,2], anti-tumor [3,4,5], anti-inflammatory [6,7], prebiotic [8], antioxidant [9], immunomodulatory [10,11], and blood glucose-lowering [12], among others. Beta-glucan are obtained from diverse sources such as plants, cereals, algae, bacteria, and fungal entities, producing highly divergent biological effects owing to disparity in physicochemical properties; namely, tertiary configuration, solubility, length of polymer, molecular weight, and degree of branching [13,14]. At present, a growing list of beta-glucan preparations has been a subject of human clinical trials [15,16].

Biosensing of beta-glucans and other pathogen associated molecular patterns (PAMPs) is primarily reserved to a constellation of germ line-encoded molecules called pathogen recognition receptors (PRRs), which include the archetypal toll-like receptors (TLRs), as well as the non-TLR receptors, namely the C-type lectin receptors (CLRs), RIG-like helicases, cytosolic DNA sensors, and NOD-like receptors (NLRs) [17,18]. Interests in dissecting the underlying molecular mechanism of beta-glucans have resulted in concerted efforts to delineate its potential cognate receptors. Early studies have principally focused on in vitro transfection experiments using cells ectopically expressing PRRs [19,20]. However, in the advent of gene targeting technologies [21], many investigators have taken advantage of the utility of beta-glucan receptor-deficient mouse models. Analysis of these gene-targeted mice has addressed the limitations imposed by using unpurified beta-glucan-containing materials such as the presence of contaminating microbial endotoxins (LPS) and, more importantly, underpins the direct involvement of the beta-glucan signaling pathway in disease resistance, progression, and susceptibility. Here, we summarized the prevailing knowledge and recent discoveries in beta-glucan research, wherein we highlighted the utility of selected gene modified mice generated by our colleagues and others, and the compelling role they played in uncovering the functional contribution of beta-glucan and its signaling cascade in fungal immunity and resistance.

## 2. Major Receptor

### 2.1. Dectin-1 and Signaling Cascade

Among the different PRRs, dectin-1 has been universally regarded as the principal receptor for beta-glucans [22,23,24,25,26]. Dectin-1 was first identified by a subtractive cloning in XS52 dendritic cell (DC) line [27] and is encoded by the *Clec7a* gene whose loci are located in a CLR gene cluster on chromosome 6 in mice and on chromosome 12 in humans [28]. It is a 43-kDa type II transmembrane glycoprotein with a single terminal CRD in the extracellular domain, which shows highly conserved structure of CLR family proteins [29], such as Dectin-2, DCIR, oxidized LDL (LOX) receptors, and asialoglycoprotein receptors (hepatic lectin 1&2) [27,30]. However, unlike others, dectin-1 has an immunoreceptor tyrosine-based activation motif (ITAM)-like motif in its intracellular domain, and lacks critical residues engaged in Ca^2+^-dependent carbohydrate binding processes [22,31]. Although the degree of concordance at the amino acid level between human and mouse dectin-1 is determined to be around 64% [30], their structural and functional properties are homologous [32].

Dectin-1 is highly expressed in lymphoid organs [31]. Contrary to earlier belief that dectin-1 is expressed almost exclusively by dendritic cells [27,33], recent investigations have elegantly elucidated that a broader variety of immune cell populations exhibit dectin-1 expression. In particular, several myeloid lineage cells such as peripheral blood neutrophils as well as peripheral and bone marrow-derived monocyte/macrophages displayed a strong dectin-1 expression, whereas there was a remarkably lower level on dendritic cells, resident peritoneal macrophages, and T cells in the spleen [26,34]. On the other hand, non-immune cells, such as alveolar epithelial cells, intestinal epithelial cells, fibroblasts, and endothelial cells, were also reported to express dectin-1 [18,22,35,36].

Drawing from the seminal work of Czop et al. [37], subsequent studies have gradually unraveled the complexities of the dectin-1 signaling pathway. Beta-glucans locate preferentially on discrete budding regions of the fungal cell wall [38,39]. Dectin-1 binds to beta-glucan with a high affinity, and then an activation signal is sent through the ITAM-like motif (hemi-ITAM) in its cytoplasmic tail, causing phosphorylation of its tyrosine residues [19,40] (Figure 1). Spleen tyrosine kinase (Syk) is then recruited to the ITAM-like motif via direct interaction between the phosphotyrosine residues of the ITAM-like sequences and the two tandem SRC homology 2 (SH2) domain of Syk [19,41]. As a consequence, Syk undergoes phosphorylation and mediates internalization of beta-glucan-enriched fungal components in DC [42,43]. Afterwards, an activation signal is then transmitted to the so-called caspase recruitment domain 9 (CARD9), a scaffold molecule that exclusively binds to B-cell lymphoma 10 (BCL10) and forms a trimolecular complex with BCL10 and mucosa associated lymphoid tissue 1 (MALT1) [44]. This complex enables bridging of the initial signaling events (beta-glucan/dectin-1/Syk) to the downstream activation of p65 and c-Rel subunits of the NF-κB transcription factor, ultimately leading to the biosynthesis of pro-inflammatory cytokines and induction of a series of anti-fungal responses [45]. Aside from this, Dectin1/Syk/CARD9 axis has also been implicated in the generation of reactive oxygen species (ROS), release of proinflammatory cytokine IL23, maturation of DC, and differential induction of Th1 and Th17 cells [32,42,46]. Alternatively, Dectin1/Syk/CARD9 axis may stimulate a noncanonical NF-κB route by acting on the subunit RelB, which negatively regulates polarization towards the Th1 and Th17 differentiation systems [47] (Figure 1B).

Downstream signal transduction of dectin-1 has been reported to be influenced by interaction with other innate immune receptors [48]. For instance, dectin-1 is able to collaborate with Toll-like receptor 2 (TLR2) to amplify the NF-κB-initiated release of cytokines like TNF-α, IL-10, IL-6, and IL-23 while it may integrate with other TLRs (TLR3, TLR7) to depress IL-12 induction in both DC and macrophages [32,49] (Figure 1C). Interestingly, dectin-1 can also promote a number of divergent signaling systems in a Syk-independent manner pathway and this involves cues emanating from a number of divergent signaling systems like Raf-1 and calcium-mediated pathways. Raf-1 kinase triggers the phosphorylation of the ser276 residue of the p65 subunit prompting RelB deactivation as it forms dimer with p65, the net outcome of which is complete abolition of the functional RelB-p52 DNA dimer [47] (Figure 1B). Since Raf-1 signaling similarly trickles down to NF-κB activation, there is a commensurate augmentation of Th1- and Th17-mediated activity. Meanwhile, dectin-1 may target a calcium-dependent system by switching on the calmodulin-dependent kinase II (CaMK)-proline-rich tyrosine kinase II (Pyk2) pathway (Figure 1C). This in turn relays signal, which harness the activation of ERK-MAPK to the elicitation of oxidative burst, stimulation of cAMP response element binding protein (CREB), and production of IL-10 cytokine [50] (Figure 1C). Recently, dectin-1 has been found to engage membrane-associated proteins such as the integral membrane protein, tetraspanins, and cooperatively induced anti-fungal and anti-cancer immunity. CD82, a member of tetraspanins also known as Kai1, colocalizes with dectin-1 and forms a cluster when dectin-1 binds beta-glucan, which promotes anti-fungal resistance through Syk-dependent release of pro-inflammatory cytokines and ROS [51] (Figure 1D) (Table 1). MS4A4A, the other member of tetraspanins characteristically expressed in tumor-associated macrophages, also co-ligates with dectin-1 and mediates tumor cytotoxicity by enhancing IFN-γ production and NK cell activation [52] (Figure 1E).

### 2.2. Dectin-1 KO (Clec7a^−/−^) Mice

In an attempt to interrogate the functional involvement of beta glucans in host defense mechanisms following fungal challenge, several exploratory studies have focused on dectin-1 as the beta-glucan cognate receptor [19,32,46,74] (Table 1). Based on these lines of reports, roles of dectin-1 in anti-fungal immunity were validated under in vivo context. Taylor et al. [24] developed a dectin-1 knockout (KO) mouse, which features a prominent ablation of the *Clec7a* gene region spanning exons 1–3. Independently, our colleagues also generated dectin-1 KO mouse by targeting exons 1 and 2 of the *Clec7a* gene [25]. Both mutant animals exhibited complete omission of dectin-1 expression in myeloid cells, and revealed that recognition of beta-glucan by dectin-1 was crucial for fungal recognition, cytokine induction, and ROS production, in consonance with earlier studies [19,33].

Assessment of dectin-1 KO mice has borne more important implications in the context of opportunistic fungal mycoses. This is accentuated by the discovery of a higher degree of human gastrointestinal candidiasis in hematopoietic stem cell transplant recipient patients who carry a single nucleotide polymorphism (SNP) in Tyr238X residue of the dectin-1 gene [75,76]. Similarly, *Candida albicans*-infected dectin-1 KO mice manifested a massive fungal dissemination in the stomach, intestines, and kidneys, and displayed a markedly abbreviated survival along with a profoundly depressed cytokine production and trafficking of inflammatory leukocytes to the peritoneum [24,77,78,79]. In converse, our colleagues [25] did not observe an increased susceptibility of dectin-1 KO mice to invasive *C. albicans* infection despite administration of a titrated fungal dose (1 × 10^6^) that is one order of magnitude higher than the lethal dose adopted in another work [24]. One logical explanation for these discrepant results is the varying strains of *C. albicans* used. Supporting this conjecture, Marakalala and co-workers [80] has verified that a strain-specific adaptive behavior governs the outcomes of fungal dissemination, cytokine production, and immune cell infiltration. This hinges on the differential changes in cell wall chitin composition as determined by the transcriptional regulation of chitin-related genes, which was found to be higher in ATCC18804 versus SC5314 strain-infected mice.

To date, the obligatory role of beta-glucans in fungal resistance has tremendously diversified beyond the common pathogenic fungi, *C. albicans*. Following infection of the pneumotropic fungi, *Pneumocystis carinii*, dectin-1 KO mice showed a significant decrease in ROS production and an overwhelming fungal invasion in the lung tissues [25]. Correspondingly, a significant repression of cytokine synthesis and leukocyte transmigration was found in *Aspergillus fumigatus*- and *Coccidioides immitis*-challenged dectin-1 KO mice [54,59]. Meanwhile, the role of dectin-1 was dispensable in deterring pathogenic fungal species such as *Cryptococcus neoformans* [53], *Histoplasma capsulatum* [81], and *Blastomyces dermatitidis* [65], suggesting that dectin-1 roles in anti-fungal immunity still have species-specificity.

Apart from fungal strain and species, it is conceivable that other factors may predict the beta-glucan-mediated host defense reaction against insulting fungi. Even though it has been recounted that mouse genetic background does not determine the response of dectin-1 KO mice to *C. albicans* infection [80], few accounts have attested otherwise. As illustrated by Carvalho et al. [82], dectin-1 KO mice on C57BL/6 (B6) background harbor a greater magnitude of fungal load in several mucosal organs in converse with the relatively unaltered tissue architecture in *C. albicans*-infected BALB/c dectin-1 KO mice. In a similar vein, B6 mice exemplified an augmented sensitivity to *Coccidioides immitis* infection in contrary to DBA/2 mice [83]. In both these investigations, the significant disparity in disease susceptibility between mice background was partly ascribed to absence of dectin-1 gene isoforms in BALB/c and DBA/2 mice genes [84,85], compared to the presence of splicing isoforms of dectin-1 in B6 mice. Interestingly, the major dectin-1 isoforms in humans have also been stratified according to alternative mRNA splicing wherein dectin-1A isoform carries a full length dectin-1 transcript while dectin-1B isoform contains an obliterated extracellular stalk region [31]. Detailed analysis of these isoforms showed that the significant degree of N-linked glycosylation in human dectin-1A is responsible for cell surface translocation enabling enhanced beta-glucan ligation and downstream activation of signaling pathways such as NF-κB, Akt, p38, and Ras-Raf-Mek1/2 [86]. Hence, dectin-1 isoforms may possibly serve as a useful determinant of host cellular responses versus fungal organisms.

Furthermore, factors such as infectious fungal load and organ sensitivity should also be considered. For example, A/J and DBA/2 inbred mice systemically challenged with 1 × 10^5^
*C. albicans* blastoconidia were able to recuperate from Candida infection but yielded to the fatal disease when given a higher dose of 3 × 10^5^ [87]. In a parallel sense, an incremental increase of *C. albicans* strain ATCC1880 inoculum up to 10 times higher than the original dose provided by our colleagues had predisposed dectin-1 KO B6 mice to a level of fungal sensitivity that is nearly tantamount to that afforded by the SC5314 strain [80].

Early experimental data in vitro have proposed that beta-glucans are deliberately exposed whenever yeast undergoes physiological budding processes [88,89]. Arguing against this concept, succeeding in vivo examinations have asserted that beta-glucans are comparably unmasked regardless of morphological types (yeast and hyphal forms) and fungal strains (SC5314 and ATCC1880) especially during the later stages of live *C. albicans* infection [80,90,91]. Considering the data generated by our colleagues, it seems likely that beta-glucans are not readily accessible for dectin-1 recognition after infection with live *C. albicans,* as shown by a decreased cytokine production of dectin-1 KO thioglycolate-stimulated macrophages. This is compatible with previous presumptions that beta-glucans are not easily extruded at the cell wall surface of live *C. albicans* but rather imbedded subjacent to the dense mannan-mannoprotein layer [92,93]. Likewise, in *Paracoccidioides brasiliensis* [94], and *Cryptococcus neoformans* [95], beta-glucans are concealed by an alpha-glucan and galactoxylomannan/glucoronoxylomannan-enriched external layer, respectively. Lending further justification to these lines of reports, treatment of live zymosan or live *C. albicans* with sodium hypochlorite (NaClO) [96], caspofungin [91] or heat-killing treatment [92,97] has dramatically improved beta-glucan exposure. However, this notion cannot be generically applied to all fungi as beta-glucans are abundantly expressed on the cell surface of live *Coccidiodes* spp. [59]. Recently, it has been shown that the proportion of beta-glucan surface expression is highly heterogeneous even at the strain level [98]. Hence, fungal organisms may variably organize beta-glucans on their cell wall surface to thwart recognition and stifle host innate immunity from mounting an appropriate anti-fungal response.

Beta-glucan recognition by dectin-1 and its downstream signaling plays important roles to tether fungal recognition to boost effector functions in the host immune system. As described above and confirmed by our colleagues using *Sparassis crispa* glucan (SCG), dectin-1-beta-glucan complex can render DCs and other antigen presenting cells to undergo maturation and activation [25]. This is followed by the release of inflammatory cytokines that can direct naive CD4^+^ T helper cells to adapt a specific differentiation program [99,100]. Where a cytokine milieu has a particular enrichment for IL-6, IL-1β, IL-23, and TGF-β may bias towards Th17 differentiation, the production of TNF-α and IL-12 may instruct cells to polarize into a Th1-type [46,101]. Activated DCs can also tailor the conversion of Treg cells into IL-17-secreting cells, which are produced by a hybrid population of human Foxp3^+^IL17^+^ T cells [102]. However, the mechanism whereby the host maintains Th1-Th17 balance is presently ill-defined [27]. Dectin-1 KO mice infected with *Coccidiodes* spp. divulged mixed Th1 and Th17 immune responses [59,103], yet neither of these adaptive differentiation modules has accorded protection in dectin-1 KO mice intraperitoneally challenged with *Trichopython rubrum* [64]. In invasive aspergillosis, dectin-1 Y238X polymorphism has prompted a decreased cytokine expression of IL-6, IL-17, IL-1β, IL-10, and IFN-γ in human PBMCs, suggesting a cooperative interrelation between Th1 and Th17 immunity [104]. Digressing from this observation, a paper by Werner and colleagues [54] has disputed this in favor of a predominant Th17 pathway, as hinted by the sizeable reduction of *A. fumigatus* clearance in dectin-1 deficient mice vis-à-vis an IL-17 blockade. Mechanistically, this perceived lineage commitment towards IL-17-producing Th17 cells has been strongly connected with a profound diminution in IFN-γ and IL-12p40 cytokine expression [56]. In human cases of mucocutaneous *C. albicans* infection, a markedly depleted IL-17 expression was specifically detected in patients bearing dectin-1 Y238X polymorphism [105]. A proclivity towards Th17 immunity was similarly documented in *C. albicans*- infected IL23p19-, IL17A- and IL17RA-deficient mice, which unveils an increased receptivity for the systemic form of candidiasis [106,107]. Combined together, an array of adaptive effector responses may be uniquely stimulated by dectin-1 depending on the offending fungal species.

The host defense mechanism may additionally opt to manipulate beta-glucan-dectin-1 signaling in light of other known bifurcations of the adaptive arm of immunity including CD8^+^ T cell-induced cytotoxicity and B cell-mediated antibody production [108]. Once activated by dectin-1, DCs can prime naive CD8^+^ T cells and bestow upon them the ability to expand and evolve into a fully differentiated cytotoxic CD8^+^ effector T cells that can competently release an apoptosis-related protein like perforins and granzymes. Activated CD4^+^ T cells, on the other hand, can prime B cells, enabling them to differentiate and secrete antibodies such as IgA and IgG [109]. Blurring these postulations, however, Drummond et al. [77] reported that dectin-1 deficiency does not alter CD8^+^ T cell proliferation and activation, while our colleagues demonstrated that antibody production is practically indistinguishable between dectin-1 WT and KO mice in the presence of activating sheep RBC or dinitrophenol-keyhole limpet hemocyanin [25]. On the grounds that the former generalizations were inferred mostly from human studies while the latter was derived largely from mouse studies, it can be reasoned that the obvious biological difference between host species may possibly address this incongruity. Also, in view of the fact that B-cell survival relies in part on signals originating from CD4^+^ T cells, defects in CD4^+^ T cell activation may precipitate B cell apoptosis and therefore create a deficit in antibody production [110].

Finally, beta-glucans may serve a pivotal role in preserving intestinal homeostasis by a delicate control of the intestinal tolerance toward commensal organisms like opportunistic fungal populations. As shown by previous studies, dectin-1- and CARD9-deficient mice had heightened susceptibility to dextran sulfate sodium (DSS)-induced colitis, because of the defective fungal killing by DCs and uncontrolled expansion of the commensal fungus, *Candida tropicalis*. [58,111]. Indeed, our colleagues have failed to detect the presence of this fungal species in their facility [61], however, when colonization of *C. tropicalis* was induced, severe inflammation of the colon in dectin-1 KO mice was observed that is consistent with those of Rahabi and colleagues’ report [112] (Figure 2).

Taken together, these accumulating lines of evidence ascertained from dectin-1 KO mice suggest that beta-glucan-dependent modulation of fungal immunity and host resistance is rather a multilayered process that is constantly shaped by a panoply of factors including fungal strain and species, genetic background of mice, specific immune cell population, fungal load, ligand type and purity, etc.

## 3. Alternative Receptors

### 3.1. CR3 KO Mice

Several alternative receptors for beta-glucans have also been described in literatures (Figure 3). Complement receptor 3 (CR3), a heterodimer comprising of αM and β2 subunits, is part of the integrin family of adhesion molecules that has been widely accepted as the principal receptor for beta-glucans long before dectin-1 discovery [113,114]. Examination of phagocytic neutrophils from CR3-deficient mice (syn. αMβ2^−/−^, CD11b^−/−^, CD18^−/−^, Mac-1^−/−^, *Itgam*^−/−^, *Itgb2*^−/−^) has unveiled the requisite role of this receptor in opsonic and non-opsonic recognition and phagocytosis of both particulate and soluble beta-glucans (Table 1). As illuminated in the study of Xia and colleagues [115] and in conformity with early human studies [113,116], the phagocytic ability of neutrophils was prominently reduced (~60%) upon exposure to serum-opsonized zymosan, whereas uptake of particulate beta-glucans was entirely abrogated in CR3 KO mice as opposed to control mice. Conflicting data exist, meanwhile, when considering other phagocytes such as macrophages, DC, and NK cells. This may occur within species where dependence for CR3 was debated to be involved in the phagocytosis of particulate beta-glucans in mice [115,117] or may arise interspecies-specifically where macrophage ingestion of unopsonized yeast particles was suggested to necessitate the recognition by CR3 receptors in humans [118,119] but not in mice [33].

Emerging evidence from human studies and CR3-deficient mice has designated a distinct niche for this particular PRR in beta-glucan research. Firstly, neutrophil-mediated anti-fungal immunity is predominantly dictated by CR3. Following induction of disseminated histoplasmosis, CR3 (*Itgam*)-deficient mice revealed a remarkable decrease in splenic levels of TNF-α, IL-17 and IFN-γ cytokines accompanied by a significant reduction of CD4^+^ and CD8^+^ T cells, increased proliferation of *H. capsulatum* and diminished SYK-JNK-AP1 signal transduction [62]. Concordantly, Techner and co-workers [120] underlined the significant depression of pro-inflammatory cytokines, enhanced fungal burden, and depressed bronchial inflammation in *A. fumigatus*-infected CR3(CD11b) KO mice. On the other hand, Li et al. [121] and another group [68] speculated that repression of CR3 substantially inhibits engulfment of *C. albicans* due to a faulty LC3B-II autophagosome accumulation in human neutrophils or due to impaired RhoGTPases exchange factors (vav1 and vav3) in neutrophils of CR3(Mac-1/CD11b) KO mice, respectively. Secondly, CR3 receptors may execute the killing of fungal hyphae or filamentous forms. In comparison to the ineffective action of dectin-1 and TLR2 against fungal hyphae [92], deletion of CR3 in CD18 KO mice neutrophils caused an impaired elimination of *A. fumigatus* and *C. albicans* hyphae coupled with a defect in calprotectin release and neutrophil extracellular trap (NET) formation [72]. Therefore, CR3 represents a promising beta-glucan receptor in neutrophils that may be exploited for anti-fungal activity.

### 3.2. EphA2 KO Mice

Just a couple of years ago, a novel PRR for beta-glucans that belongs to the receptor tyrosine kinase (RTK) superfamily had been introduced—the ephrin type-A receptor 2 (EphA2) [73,122] (Figure 3). EphA2 has been known to be involved in various physiological functions such as cell–cell communication, cell growth and differentiation, and cell migration [123,124], as well as in prostate, mammary, esophageal, cervical, lung, ovarian, renal cancers and leukemia [125,126,127,128,129,130,131,132], and inflammatory conditions such as ischemia-reperfusion, psoriasis, atherosclerosis, and acute lung injury [133,134]. In the recent breakthrough study by Swidergall et al. [73], it has been characterized as an epithelial-type of PRR in the oral epithelium that fulfills an incredulous function in host defense against oropharyngeal candidiasis (OPC) (Table 1). *C. albicans*-infected EphA2-deficient mice disclosed significantly undermined early anti-fungal responses, with decreased levels of various chemokines such as CCL3 and CXCL1/KC, and cytokines like IL-1β, IFN-γ, and IL-17A. An inquiry into the underlying mechanism incriminates the stringent control exerted by EphA2 signaling on two signal transduction pathways, the mitogen associated protein kinase (MAPK) and signal transducer and activator of transcription 3 (STAT3), both of which are considered critical in oral mucosal fungal sensing and immunity [135,136].

EphA2 has also been known to play diverse roles in anti-fungal immunity. Its relevance in oral fungal resistance has been exemplified by the sharp reduction in the frequency of recruited neutrophils and inflammatory monocytes in EphA2 KO mice relative to WT animals [137]. Succeeding experiments showed that EphA2 in neutrophils but not in inflammatory monocytes is vital in conferring resistance against serum-opsonized *C. albicans* yeast during the restricted early period of infection. Following contact of EphA2 receptors to serum-opsonized yeast, the p40^phox^ cytosolic protein of the NADPH oxidase system is concomitantly phosphorylated, eventually stimulating ROS-mediated fungal killing activity [137]. Similar to what has been described in dectin-1 receptors, EphA2 does not participate in the clearance of invasive fungal hyphae, but binds with other fungal species (*C. glabrata, A. fumigatus, Rhizopus delemar*), and collaborates with other PRR, specifically the epidermal growth factor receptor (EGFR), which governs the uptake of pervading fungal pathogens and sustains EphA2 signaling by stabilizing receptor phosphorylation [73,138]. Collectively, this series of information, together with the limited contribution of classical beta-glucan PRRs in addressing oropharyngeal infection [139], indicate the compelling role of EphA2 in holding the fort in oral mucosal immunity.

Other known receptors for beta-glucans include lactosylceramide (LacCer) [140]. Currently, information is scarce regarding its precise role in beta-glucan function and signaling.

## 4. Closing Remarks

The utility of various beta-glucan receptor deficient mouse models during the last couple of decades has shed important light and answered mystifying conundrums concerning the composite roles of beta-glucans in various physiological processes and diseases. Although the bulk of published data have been concentrated on its role in fungal immunity, there are still unknown aspects to be learnt and decoded in this area. As new insights have gradually becoming available, especially on previously unexplored and understudied facets of beta glucan research, and as novel beta-glucan receptors are continually being elucidated, these receptor-deficient animals will stay relevant in the coming years and command more enthusiasm especially from scientists and researchers who are advancing the field of translational medicine.

## Figures and Tables

**Figure 1 ijms-22-04778-f001:**
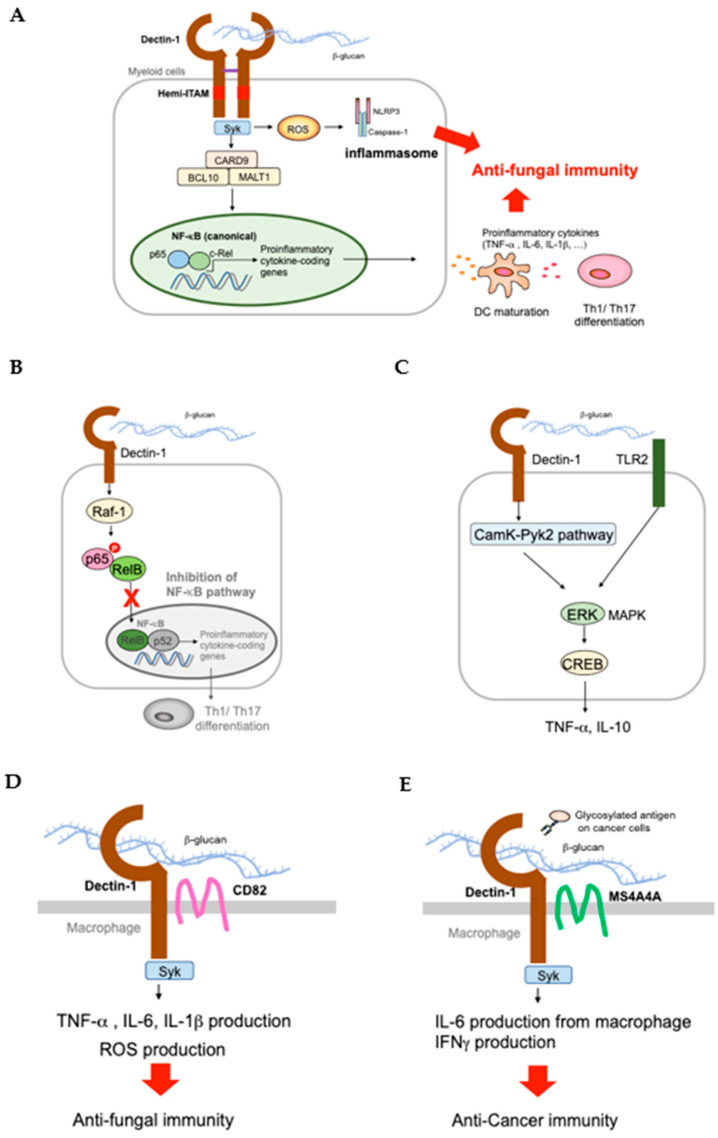
Dectin-1 Signaling Pathway. (**A**) Upon beta-glucan binding, dectin-1 recruits Syk to the hemi-ITAM followed by the activation of the CARD9-BCL10-MALT1 (CBM) complex. As a result, the NF-kB pathway is activated, and genes coding proinflammatory cytokines (TNF-α, IL-6, IL-1β), are released to promote the maturation of dendritic cells, and differentiation of Th1/Th17 cells. ROS production is also induced in a Syk-dependent manner, resulting in inflammasome and activation of caspase-1. All these downstream events of dectin-1 finally contribute to the eradication of fungi. (**B**) Raf-1 is activated in a Syk-independent manner and induces deactivation of RelB and activation of NF-kB pathway. (**C**) Dectin-1 collaborates with calcium-mediated pathways to activate ERK-MAPK, cAMP response element binding protein (CREB), and induce cytokines including TNF-α, and IL-10. (**D**) Dectin-1 collaborates with tetraspanin-like molecules, binds beta-glucan, and transduces signals. (**E**) CD82 colocalize with dectin-1 during beta-glucan binding, and induces inflammatory cytokines and ROS to boost anti-fungal immunity. MS4A4A also binds to dectin-1 when dectin-1 binds beta-glucan or glycoprotein on cancer cells, and enhances IL-6 cytokine production from macrophages, IFN-γ production, and contributes to anti-cancer immunity.

**Figure 2 ijms-22-04778-f002:**
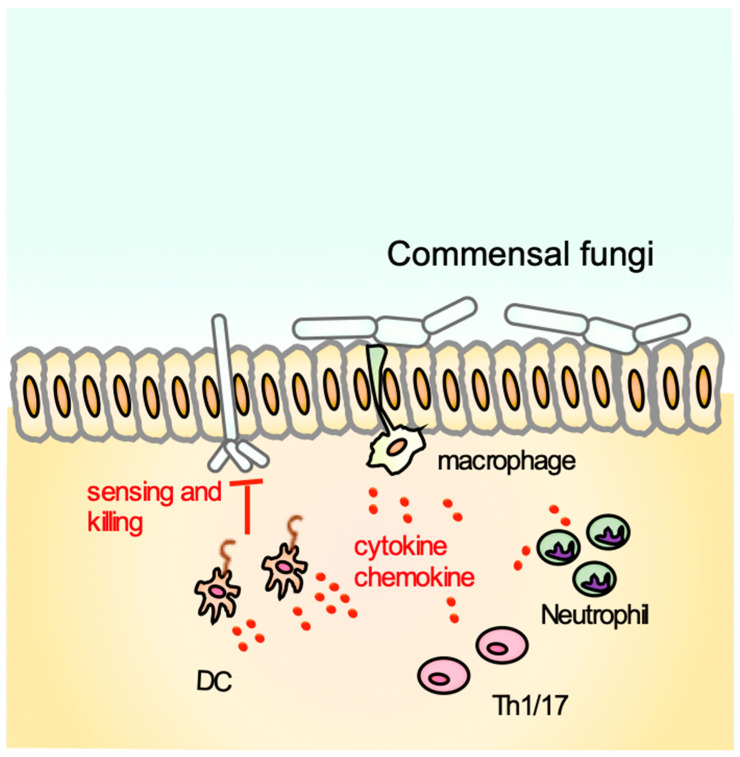
Beta-glucans may regulate intestinal homeostasis. Presence of the commensal fungi, *Candida tropicalis* enhances the susceptibility of dectin-1 KO mice to DSS-induced colitis by impairing the fungal killing activity of DCs and macrophages, and further activation of T cells, neutrophils by cytokines and chemokines from DCs and macrophages. This in turn enhances further growth of *Candida tropicalis* [58,111].

**Figure 3 ijms-22-04778-f003:**
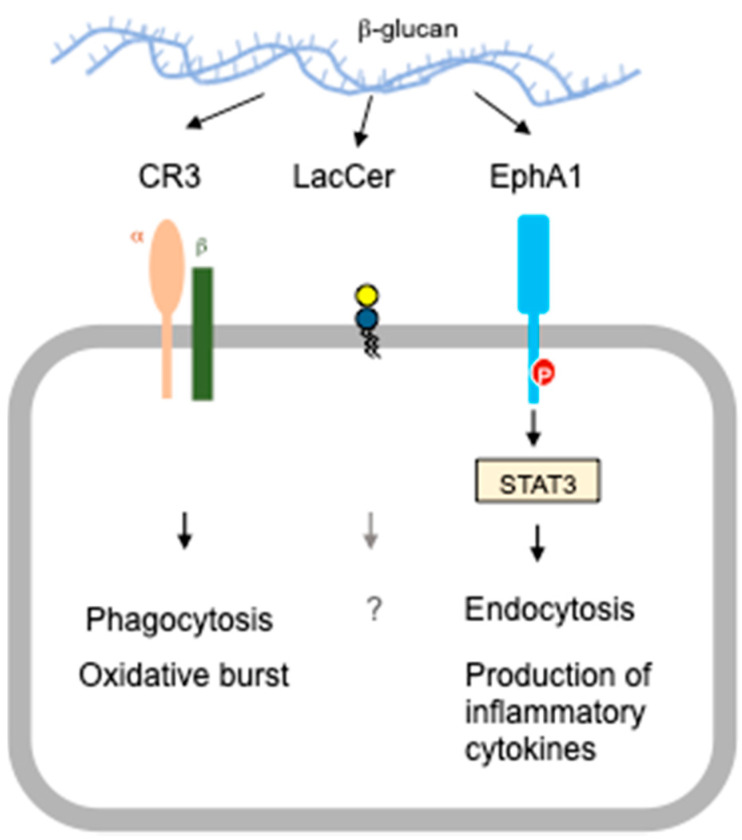
Alternative receptors of β-glucan. CR3 (hetero-complex of CD11b/Integrin αM and CD18/Integrin β2) and EphA2 are known to induce phagocytosis of beta-glucan particles or fungi, oxidative burst, and production of inflammatory cytokines, suggesting that they are functional receptors for beta-glucan in anti-fungal immunity. LacCer is also known to bind beta-glucan, however, its precise function in beta-glucan signaling is presently unknown.

**Table 1 ijms-22-04778-t001:** In vivo studies of fungal infection using beta-glucan receptor (related) gene-deficient mice.

Receptor	Allele Symbol	Genetic Background	Fungus	Journal	Ref.
Dectin-1	*Clec7a* <tm1Yiw>	C57BL/6J, BALB/c	*Candida albicans, Pneumocystis carinii*	Nat Immunol. 2007	[25]
	*Clec7a* <tm1Gdb>	B6;129 mix	*C.albicans*	Nat Immunol. 2007	[24]
	*Clec7a* <tm1Yiw>	C57BL/6J	*Cryptococcus neoformans*	Microbiol Immunol. 2007	[53]
	*Clec7a* <tm1Gdb>	129/SvEv	*Aspergillus fumigatus*	J Immunol. 2009	[54]
	*Lyz2* <tm1(cre)Ifo>. *Clec7a* <tm1.1Bpip>: *	C57BL/6J	*C.albicans*	PLoS Pathog. 2010	[55]
	*Clec7a* <tm1Yiw>	C57BL/6J	*A. fumigatus*	J Exp Med. 2011	[56]
	*Clec7a* <tm1Yiw>	BALB/c	*A. fumigatus*	PLoS One 2011	[57]
	*Clec7a* <tm1Gdb>	C57BL/6	*Candida tropicalis*	Science 2012	[58]
	*Clec7a* <tm1Gdb>	C57BL/6, (C57BL/6;DBA/2)F2	*Coccidioides immitis*	mBio 2013	[59]
	*Clec7a* <tm1Gdb>	C57BL/6	*Paracoccidioides brasiliensis*	J Infect Dis. 2014	[60]
	*Clec7a* <tm1Yiw>	C57BL/6J	*C. tropicalis*	Cell Host Microbe 2015	[61]
	*Clec7a* <tm1Gdb>, *Clec7a* <tm1Gdb>. *Itgam* <tm1Myd>	C57BL/6	*Histoplasma capsulatum*	PLoS Pathog. 2015	[62]
	*Clec7a* <tm1Yiw>	C57BL/6J	*Trichosporon asahii*	Inflamm Res. 2016	[63]
	*Clec7a* <tm1Yiw>	C57BL/6J	*Trichophyton rubrum*	Innate Immun. 2016	[64]
	*Clec7a* <tmX>: **	C57BL/6	*Blastomyces dermatitidis*	J Clin Invest. 2016	[65]
	*Clec7a* <tm1Gdb>	C57BL/6	*C.albicance, Candida krusei*	Am J Transl Res. 2019	[66]
	*Itgam* <tm1Myd>	C57BL/6	*B. dermatitidis*	J Immunol. 2004	[67]
	*Itgam* <tm1Myd>	C57BL/6	*C.albicans*	Cell Host Microbe. 2011	[68]
	*Itgam* <tm1Bll>	C57BL/6J	*C.albicans*	Infect Immun. 2011	[69]
CR3	*Itgb2* <tm2Bay>	C57BL/6J	*A. fumigatus*	J Clin Invest. 2012	[70]
	*Itgam* <tm1Myd>, *Clec7a* <tm1Gdb>. *Itgam* <tm1Myd>	C57BL/6	*H. capsulatum*	PLoS Pathog. 2015	[62]
	*Itgam* <tm1Myd>	C57BL/6	*C. neoformans*	Nat Commun. 2019	[71]
	*Itgam* <tm1Myd>	C57BL/6	*A. fumigatus*	Front Immunol. 2019	[72]
EphA2	*Epha2* <tm1Jrui>	C57BL/6	*C.albicans*	Nat Microbiol. 2018	[73]
CD82	*Cd82* <tm1.1Cmir>	C57BL/6	*C.albicans*	J Immunol. 2019	[51]

* Macrophage-specific conditional knock-out. ** Undescribed.

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
