# Peer review of "Insights on the Functional Role of Beta-Glucans in Fungal Immunity Using Receptor-Deficient Mouse Models"

_ijms, 2021, doi:10.3390/ijms22094778_

Round 1

Reviewer 1 Report

This submission reviews the functional role of beta-glucans using receptor-deficient mouse models. The length of this manuscript needs to be reduced to avoid boring the readers, particularly the length of sections 2.1 and 2.2 . Instead of using many English words not very meaningful to readers, the authors should be carefully about the conclusion they made. For example, why the authors used only one reference [22] to conclude that dectin-1 is the principal receptor for beta-glucans? A more reasonable approach is to include experimental data reported by at least three different labs. The abstract also needs revision to let the readers clearly catch the main part of this review. The authors can use one or two sentenses summarizing each figure to make the abstract. An entensive English editing is required before acceptance of this review and this paper should be edited to remove many unnecessary descriptive terms and just keep sentences describing novel findings and important conclusions. The title may be changed to : Insights on the functional role of beta-glucans in fungal immunity using  receptor-deficient mouse models, since it seems that beta-glucan-mediated fungal immunity is the core of this review.

Author Response

  1. This submission reviews the functional role of beta-glucans using receptor-deficient mouse models. The length of this manuscript needs to be reduced to avoid boring the readers, particularly the length of sections 2.1 and 2.2 .

Response: The length of the manuscript is reduced as per recommendation. In particular, section 2.1 is renamed “2.2” in the revised manuscript and reduced from 5 pages to ~3 pages as shown in page 5 to line 14 of page 7. For section 2.2, we have decided to remove this section and retain only a small portion that has relevance to fungal immunity (as shown in page 7, lines 30-39) to make the present manuscript more focused.

  1. Instead of using many English words not very meaningful to readers, the authors should be carefully about the conclusion they made. For example, why the authors used only one reference [22] to conclude that dectin-1 is the principal receptor for beta-glucans? A more reasonable approach is to include experimental data reported by at least three different labs.

Response: We have tried to simplify the manuscript and deleted many descriptive phrases as can be seen throughout the revised manuscript. Concerning the above-mentioned example (reference 22), we have revised accordingly as shown by the inclusion of 4 more references in page 2, line 15 (reference 23-26). Moreover, we have also substituted terms or vocabularies that are unconventional and not easily comprehensible to facilitate better communication of information. For example, the word “mollified” is changed to “addressed” in page 2, line 4 of the revised manuscript. Other similar revisions are highlighted in red throughout the revised manuscript.

  1. The abstract also needs revision to let the readers clearly catch the main part of this review. The authors can use one or two sentenses summarizing each figure to make the abstract.

Response: The abstract is reconstructed to make it more appealing. It features the salient points of the major sections of the paper as shown in page 1, lines 13-24 of the revised manuscript.

  1. An entensive English editing is required before acceptance of this review and this paper should be edited to remove many unnecessary descriptive terms and just keep sentences describing novel findings and important conclusions.

Response: As mentioned, we have tried to simplify the manuscript and deleted many descriptive phrases that unnecessarily lengthen the paper. These substantial changes are reflected throughout the revised manuscript.

  1. The title may be changed to : Insights on the functional role of beta-glucans in fungal immunity using  receptor-deficient mouse models, since it seems that beta-glucan-mediated fungal immunity is the core of this review.

Response: As recommended, we have changed the title of the present review article as shown in page 1, lines 2-3.

Reviewer 2 Report

In this review, the authors provide a comprehensive evaluation of beta-glucan signaling and the role of the various beta-glucan receptors based on data generated by various groups in KO mouse models. As indicated, the review is very comprehensive, although clearly focused on Dectin-1 for which more data are available and with a strong focus toward involvement in fungal infections. I have the following suggestions:

  1. Scope of the review: Most of the data are clearly toward fungal infections, and there is a large part of beta-glucan field (e.g., cereals and nutrition, bacterial infections) that is not covered at the same depth. I would recommend that the authors limit their review to the role of these receptors in fungal infection, since that will help the review being slightly more focused and will benefit a more targeted audience. As such, I would also focus the title and abstract on fungal infections.
  2. In several places the authors conclude a section with a concluding statement. This is quite useful. I would recommend that the authors break their section 2 into more subsections with a concluding paragraph for each of these subsections. That will help the readers.
  3. The review is well written; however, the style is rather different from typical scientific English with the use of frequent “out-of-date or out-of-style” vocabulary (e.g., mollify, congruent to name a couple). While this is not incorrect, this may present a barrier and some potential misunderstanding for some readers who may not be proficient in English. I would recommend using simpler and more current vocabulary.
  4. Line 76: I’d use “lymphoid organs” rather than “lymphatic organs”.
  5. Lines 205-206: what is meant by “which is not associated with a defective bile acid synthesis”? I could not understand the relevance of this additional qualifier.
  6. Please review the figure legends.  In some places, more information is needed (e.g., Fig. 3), so they can stand on their own.  Also, the Greek symbols are lost, and there are several typos (e.g. for Fig.1, “As a result”, Dectin-1 instead of Dection, etc).

Author Response

  1. Scope of the review: Most of the data are clearly toward fungal infections, and there is a large part of beta-glucan field (e.g., cereals and nutrition, bacterial infections) that is not covered at the same depth. I would recommend that the authors limit their review to the role of these receptors in fungal infection, since that will help the review being slightly more focused and will benefit a more targeted audience. As such, I would also focus the title and abstract on fungal infections.

Response: The scope of the review article is revised to concentrate on the role of beta-glucan in fungal immunity by deleting or concentrating other topics. Only a small portion of section 2.2 that has relevance to fungal immunity in the revised manuscript  was retained (page 7, lines 27-35). The title and abstract is also revised accordingly as can be seen in page 1, lines 2-24 of the revised manuscript.

  1. In several places the authors conclude a section with a concluding statement. This is quite useful. I would recommend that the authors break their section 2 into more subsections with a concluding paragraph for each of these subsections. That will help the readers.

Response: We have tried to break the paragraphs into subsections as shown in page 5, lines, 37-39, and added a concluding statement as shown in page 7, lines 7-9; and page 9, lines 36-37.

  1. The review is well written; however, the style is rather different from typical scientific English with the use of frequent “out-of-date or out-of-style” vocabulary (e.g., mollify, congruent to name a couple). While this is not incorrect, this may present a barrier and some potential

Response: We have substituted terms or vocabularies that are unconventional and not easily comprehensible to facilitate better communication of information. For example, the word “mollified” is changed to “addressed” in page 2, line 5, “buttress” to promotes” in page 3 line 23, and “congruently” to interestingly” in page 5, line 50 of the revised manuscript. Other similar revisions are highlighted in red as can be seen throughout the revised manuscript.

  1. Line 76: I’d use “lymphoid organs” rather than “lymphatic organs”.

Response: The term “lymphatic organs” is changed to “lymphoid organs” in page 2 line 27.

  1. Lines 205-206: what is meant by “which is not associated with a defective bile acid synthesis”? I could not understand the relevance of this additional qualifier.

Response: This particular phrase is deleted and the entire paragraph is refined as shown in page 5, lines 13-20.

  1. Please review the figure legends.  In some places, more information is needed (e.g., Fig. 3), so they can stand on their own.  Also, the Greek symbols are lost, and there are several typos (e.g. for Fig.1, “As a result”, Dectin-1 instead of Dection, etc).

Response: Figure legends are revised as shown in page 4, page 8, line 5 and page 10, line 35 of the revised manuscript.

Round 2

Reviewer 1 Report

The revised manuscript now is much more readable than its original version, as many unnecessary desriptive terms have been deleted. I recommend the acceptance of this manuscript, but the authors need to check if every sentence in the revised version is logically connected to each other before a final acceptance of this article.